# Variability in Bacteriophage and Antibiotic Sensitivity in Serial *Pseudomonas aeruginosa* Isolates from Cystic Fibrosis Airway Cultures over 12 Months

**DOI:** 10.3390/microorganisms9030660

**Published:** 2021-03-22

**Authors:** Isaac Martin, Dervla T. D. Kenna, Sandra Morales, Eric W. F. W. Alton, Jane C. Davies

**Affiliations:** 1National Heart and Lung Institute, Imperial College London, Emmanuel Kaye Building, London SW3 6LY, UK; e.alton@imperial.ac.uk (E.W.F.W.A.); j.c.davies@imperial.ac.uk (J.C.D.); 2Royal Brompton & Harefield Hospitals, Sydney St., London SW3 6NP, UK; 3Antimicrobial Resistance and Healthcare Associated Infections (AMRHAI) Reference Unit, National Infection Service, Public Health England, 61 Colindale Avenue, London NW9 5EQ, UK; dervla.kenna@phe.gov.uk; 4AmpliPhi Biosciences Corporation, Global Research, Brookvale, NSW 2100, Australia; morales.sandra@gmail.com

**Keywords:** bacteriophage, *Pseudomonas aeruginosa*, antimicrobial resistance, cystic fibrosis, novel antimicrobials, adjunctive therapy, pulmonary infection

## Abstract

Antibiotic treatment for *Pseudomonas aeruginosa* (Pa) in cystic fibrosis is limited in efficacy and may lead to multi-drug resistance (MDR). Alternatives such as bacteriophages are being explored but well designed, and controlled trials are crucial. The rational selection of patients with bacteriophage susceptible infections is required for both safety and efficacy monitoring. We questioned whether bacteriophage susceptibility profiles were constant or variable over time, variability having been reported with antibiotics. Serial Pa isolates (n = 102) from 24 chronically infected cystic fibrosis (CF) patients over one year were investigated with plaque and antibiotic disc diffusion assays. Variable number tandem repeat (VNTR) analysis identified those patients with >1 isolate. A median (range) of 4 (3–6) isolates/patient were studied. Twenty-one (87.5%) individuals had a single VNTR type; three (12.5%) had two VNTR types at different times. Seventy-five percent of isolates were sensitive to bacteriophage at ≥ 1 concentration; 50% of isolates were antibiotic multidrug resistant. Serial isolates, even when representing a single VNTR type, varied in sensitivity to both bacteriophages and antibiotics. The rates of sensitivity to bacteriophage supports the development of this therapy; however, the variability in response has implications for the selection of patients in future trials which must be on the basis of current, not past, isolate testing.

## 1. Introduction

Cystic fibrosis (CF) is an autosomal recessive condition in which affected individuals are at an increased risk of bacterial lung infection with opportunistic pathogens. Repeated and chronic airway infections result in a decline in lung function, decreased quality of life and increased mortality. Median predicted age of survival for people with CF in the UK is 47 years according to projections in 2017, while the median age of death was 31 for the same year [1].

*Pseudomonas aeruginosa* (Pa) is the pathogen leading to the greatest burden of lung disease in CF, as a majority of patients are infected with this ubiquitous Gram-negative bacterium by the time they reach adulthood. Data from the 2017 UK CF registry show that 5.4% of the paediatric population (<16 years) and 44.5% of adults ≥16 years of age are chronically infected with Pa [1]. The natural history of infection is typically through initial acquisition of the bacteria, to intermittent infection, and eventually to chronic colonization [2,3]. Though nosocomial infection with epidemic strains—often associated with increased rates of antimicrobial resistance (AMR)—is well documented [4,5,6], early infection is usually with a “wild-type” organism that has not yet adapted to the lung environment and is therefore more amenable to standard antibiotic treatment [3] and early eradication [7,8]. If, however, these measures prove unsuccessful, several factors contribute to changes that make the bacteria a chronic, host-adapted pathogen.

The organism has a number of inherent and acquired characteristics which can confer resistance to treatment and allow it to persist in the lungs. One such change is the evolution of the mucoid phenotype—through the copious production of alginate, an exopolysaccharide which provides a thick, sticky barrier to antimicrobials and host defences [9]. Such phenotypic change is a survival mechanism that can be driven by factors inherent in the CF lung [9,10], but also by external factors aimed at its eradication, such as antibiotic administration [11].

The cornerstone of treatment of infection in CF is antibiotics, to which improvements in quality of life as well as longevity have been widely attributed [12]. However, rates of antimicrobial resistance (AMR) are high [13], limiting the drugs armoury. AMR is an obstacle to effective treatment, has been associated with poorer patient outcomes [14] and is increasing over time based on registry data [15]. However, even without the restrictions of AMR, studies looking at the antibiotic choice based on traditional antibiotic sensitivity testing show poor/no correlation with clinical outcome [16,17], a fact which has led many to explore new treatments and sensitivity testing methods.

Bacteriophages (or phages) are viruses that bind to and infect specific bacterial cells. Virulent (lytic) phages replicate within their host, leading to the lysis of the bacterium and the release of viral progeny which can, in turn, infect and lyse other bacteria. In response to the threat of AMR, there has been a resurgence of interest in phages to tackle Pa [18] and other difficult-to-treat infections [19]. We have previously reported in a murine model that antipseudomonal bacteriophages resulted in a reduction in both bacterial load and inflammatory response [20]. In the clinical sphere, support for bacteriophage treatment for patients with CF is largely anecdotal [19,21] and rigorously conducted clinical trials are urgently needed. However, several questions remain around study design and patient selection. It is intuitive that clinical efficacy will require patients with phage-sensitive organisms; heterogeneity in sensitivity is clearly likely to dilute the efficacy signal. In addition, however, any clinical safety signal related to bacterial lysis would be missed in those harbouring phage-resistant strains. To aid future patient selection, we have, therefore, determined whether bacteriophage sensitivity patterns are stable or variable over time. We hypothesised that, as has been described for conventional antibiotics, longitudinal, intra-subject variability in bacteriophage susceptibility patterns would occur. The confirmation of this would help underpin selection criteria for recruitment to future studies.

## 2. Materials and Methods

### 2.1. Pseudomonas aeruginosa Isolates

Following culture and matrix-assisted laser desorption/ionisation time-of-flight mass spectrometry (MALDI-TOF–MS) [22] confirmation through our site’s clinical service, Pa strains isolated from airway samples (sputum, cough/throat swabs, bronchoalveolar lavage) were stored in a research repository (−80 °C on glass Microbank™ beads; Pro-Lab Diagnostics, Richmond Hill, ON, Canada). All isolates had been visually categorized by a hospital microbiologist as either mucoid or non-mucoid. For this study, isolates were chosen from 24 individuals who had 4 or more positive Pa cultures ≥1 month apart over ≥8 months of 2016. Bacteria were grown overnight at 37 °C on nutrient agar (Oxoid, ThermoFisher Scientific, Basingstoke, RG24 8PW, U.K.). CF patients had consented to their clinical data being entered into the national patient registry and being used for approved research projects. Isolates and data were coded and pseudonymized for the study.

### 2.2. Antibiotic Sensitivity Testing

Antibiotic sensitivity testing was performed by disc diffusion assay according to the European Committee on Antimicrobial Susceptibility Testing (EUCAST) protocols [23]. *Pseudomonas aeruginosa* ATCC ^®^ 27853™, a reference strain with established sensitivity cut-offs was used as a quality control measure for each group of disc diffusion assay testing. A widely accepted definition of AMR (resistance to ≥1 antibiotic in ≥3 antibiotic classes) [24] was applied. We tested 9 antibiotics in 4 classes: aminoglycosides (amikacin, gentamicin, tobramycin), β-lactams (aztreonam, ceftazidime, meropenem, piperacillin-tazobactam), fluoroquinolones (ciprofloxacin) and sulfonamides (co-trimoxazole).

### 2.3. Bacteriophages and Plaque Assays

A cocktail of 4 lytic phages (Pa193, Pa204, Pa222 and Pa223 each at a calculated titre of 1 × 10^9^ [21,25] in Tris-buffered salt-magnesium buffer (SMB) was provided by AmpliPhi Australia. The final concentration of the neat solution was a titre of 4 × 10^9^. Previous work (personal communication) had confirmed activity of the cocktail in plaque assay against >80 of 100 Pa isolates from a historical CF sample set from the Royal Brompton Hospital (RBH).

After overnight culture in nutrient broth (Oxoid), each Pa isolate was diluted in fresh nutrient broth to an optical density (OD) of 0.1 (~2 × 10^7^ CFU/mL) [26]. One hundred microlitres (100 µl) of this suspension was mixed with 3 mL of semi-solid nutrient agar (13 g/L Oxoid Nutrient Broth + 4 g/L Oxoid Agar), poured over sterile nutrient agar plates (13 g/L Oxoid Nutrient Broth + 15 g/L Oxoid Agar) and allowed to cool for 20–30 min. Five microlitre (5 µL)-aliquots of serial log_10_ dilutions of phage cocktail in SMB (neat to 10^−6^) were spotted onto the surface of the Pa-inoculated plates, each assay being performed in triplicate. Five microlitres (5 µL) of SMB was used as a negative control. Once dry, inverted plates were incubated at 37 °C for 16–24 h.

Agar plates were examined over a dark background. If there were no discernible plaques (zones of clearance in the bacterial lawn) at any dilution, the phage cocktail was scored as no lytic effect. If plaques were observed, the apparent titre in plaque forming units per millilitre (PFU/mL) was calculated using the most dilute solution at which plaques were observed according to the following formula:

Apparent phage titre = [(PFU in spot)/(spot volume in mL)] × 1/(dilution factor of tube from which spot was taken)

We chose two definitions of susceptible: lytic activity with neat phage (corresponding to an apparent titre of 2 × 10^2^ PFU/mL) or at thousand-fold dilution (corresponding to an apparent titre of 2 × 10^5^ PFU/mL).

### 2.4. Longitudinal Variability in Antibiotic and Phage Susceptibility

Each patient was classified as having either a stable (always resistant or always sensitive) or variable response to each antibiotic agent and to the phage cocktail. For each agent, the percentages of patients showing a stable or variable patterns are presented.

### 2.5. Variable Number Tandem Repeat (VNTR) Typing

Variable number tandem repeat (VNTR) analysis was performed on isolates by Public Health England [27]. Nine loci were used for strain typing according to the above method (ms61, ms172, ms207, ms209, ms211, ms214, ms217, ms222, and ms213). The Liverpool strain (NCTC 13415) was used as a control in each batch of experiments.

### 2.6. Lung Function Tests

Data were obtained from the UK CF Patient Registry and hospital electronic records. For the 23 adults in this dataset, European Coal and Steele reference equations had been used to determine the % of predicted forced expiratory volume in 1 s (FEV_1_) [28]. The one paediatric patient whose isolates were used in this dataset used the Rosenthal equation as reference for determination of FEV_1_ % predicted [29].

### 2.7. Statistical Analysis and Graphical Representation

All statistical analysis and graphs were generated using Prism 8.0 (GraphPad, San Diego, CA, USA). For parametric data, standard *t*-tests were used to compare means. A Mann–Whitney *U* test was used to compare median proportions of isolates with variable responses to antimicrobials for those with more than one Pa strain to those with one strain. Patient level data were also analysed using a Fisher’s exact test with sensitivity data converted to categorical outcomes (sensitive vs. resistant).

## 3. Results

### 3.1. Patient Demographics and Isolate Characteristics

Baseline patient characteristics, including FEV_1_, VNTR results, and Pa isolate characteristics (including multi-drug resistance (MDR) status) are summarized in Table 1. From the 24 patients, a total of 112 isolates were selected for testing (median (range) 4 (3–6)/subject), of which 102 (91%) had antibiotic/bacteriophage sensitivity profiles and VNTR data. Isolates without VNTR data (10) were excluded from the final analysis.

On the basis of VNTR, 21 (87.5%) patients had a single genetic isolate obtained on all their serial samples. In three patients (12.5%), there were two distinct VNTR profiles obtained. One (subject F) had two isolates of the same strain from early 2016, and two further isolates from another later in the year, each with distinct antibiograms and phage sensitivity profiles. In two subjects (K and L), one isolate of four and three samples, respectively, differed extensively from the others (Appendix A). In terms of the stability/variability of responses to bacteriophage and antibiotics, we considered data from these two groups separately, although the small n in the latter limits inter-group comparisons. Most subjects had unique strains; the only strains common to more than one subject were the Liverpool Epidemic Strain (LES—four patients) and Cluster A (two patients) [30].

### 3.2. Antibiogram Profiles and Phage Susceptibility

**Table 2 microorganisms-09-00660-t002:** Inter-patient antibiogram and plaque assay variability.

	*N* patients			
Antimicrobial	Stably Sensitive	Variable	Stably Resistant	Stable/Variable (%)
Bacteriophage (neat cocktail)	10 (*1)	10 (*2)	1	52/48 (*33/67)
Bacteriophage (10**^−^**^3^ dilution)	1	17 (*3)	3	19/81 (*0/100)
Amikacin	6	9 (*2)	8 (*1)	67/33 (*33/67)
Aztreonam	8(*1)	6 (*2)	7	71/29 (*33/67)
Ceftazidime	9	6 (*3)	6	71/29 (*0/100)
Ciprofloxacin	1	9 (*3)	11	57/43 (*0/100)
Gentamicin	4	9 (*2*	8 (*1)	57/43 (*33/67)
Meropenem	8	6 (*2)	7 (*1)	71/29 (*33/67)
Piperacillin-Tazobactam	10	6 (*2)	5 (*1)	71/29 (*33/67)
Tobramycin	14 (*1)	3 (*2)	4	86/14% (*33/67)

Numbers are presented for patients with a single VNTR strain and asterisked * for those with >1 Pa strain, which were shown in parentheses to demonstrate the spread of data and have been presented separately in the tallies in the final column. There is no European Committee on Antimicrobial Susceptibility Testing (EUCAST) cut-off for co-trimoxazole as it is not a standard anti-pseudomonal antibiotic; the results are therefore not presented here.

There was a high degree of variability in the sensitivity of strains to the bacteriophage cocktail. Firstly, amongst the 21 subjects with only a single VNTR strain over the course of the study: 10 patients (48%) harboured Pa strains that demonstrated stable sensitivity to the neat phage (phage titre > 2 × 10^2^ PFU/mL); one patient (5%) had stably resistant isolates. If applying “susceptibility” only to those isolates demonstrating lytic activity at the lower phage concentration of 10^−3^ dilution (phage titre > 2 × 10^5^ PFU/mL), greater variability was seen. Specifically, only one patient’s isolates (5%) were stably sensitive, those from the majority (17 patients, 81%) were variable, whilst stable resistance was seen in three patients (14%); Table 2. As expected, patients were much more likely to be sensitive to the neat phage preparation than to the 10^−3^ dilution (*p* = 0.004, using Fisher’s exact test).

For antibiotics, there was also a high degree of variability in the disc diffusion antibiogram results, even in those with a single Pa strain confirmed on VNTR. Within an individual, susceptibility to tobramycin was most consistent and to ciprofloxacin most variable (Table 2). Although it was not possible to correlate individual isolates with the time courses of antibiotics in this complex patient group, 18/24 (75%) of this group received ciprofloxacin, while 22/24 (92%) received tobramycin over the time period of this study.

With both phage and antibiotics, the isolates demonstrated high levels of variability over the year. Data from multiple antibiotics allowed us to compare a median proportion of isolates demonstrating variable responses to drugs between subjects with two VNTR types or only one; as might be expected, variability was higher in the former (67%), but was substantial even in the group with a single strain (29%; *p* < 0.001; Mann–Witney *U*).

Figure 1 shows example antibiograms and plaque assay results for four patients chosen to illustrate different sensitivity patterns. The first three of these have a single VNTR type (H: stable Abx, stable phage; B: substantially variable Abx, variable phage; D: subtly variable Abx, variable phage). Thus, even when all isolates from a patient possessed the same VNTR type, susceptibility to antibiotics, phage or both could be variable. The fourth subject had two VNTR types (F: variable Abx, variable phage). Much of the subject F’s variability appears to relate to these strain differences. There were two other patients (K and L) in whom more than one strain was identified (VNTR data in the Appendix A); in these subjects, variability mapped less clearly to the strain types (example patient K, Figure 2b).

### 3.3. Similarity of Results in Patients Sharing a Common Pa Strain

Four patients were infected with the LES, which was the most commonly shared strain. Two other patients were infected with Cluster A, a strain which often shows no epidemiological link, but has been isolated from hospital and environmental sources across the UK [30].

**Figure 2 microorganisms-09-00660-f002:**
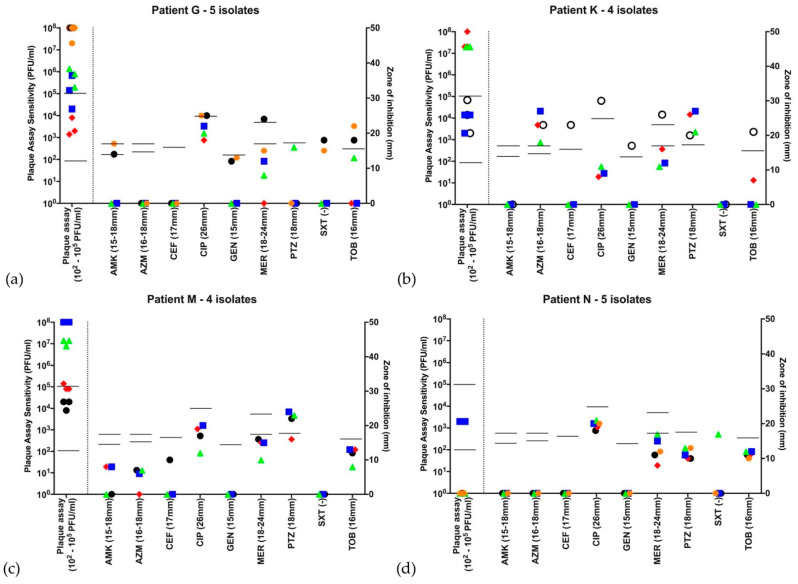
Antibiograms and bacteriophage sensitivity profiles for 4 patients with the Liverpool Epidemic Strain. Isolates are graphically represented by different symbols chronologically throughout the year. The first isolate is represented by 
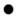
, with subsequent isolates 
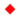
, 
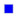
, 
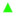
, 
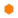
, if there were 5 isolates through the course of 2016. Open symbols represent different strains, identified by VNTR analysis. Abbreviations—AMK: amikacin; AZM: aztreonam; CEF: ceftazidime; CIP: ciprofloxacin; GEN: gentamicin; MER: meropenem; PTZ: piperacillin-tazobactam; SXT: co-trimoxazole (no EUCAST breakpoint); TOB: tobramycin. Disc diffusion assay for antibiotic sensitivity testing was performed once at each time point and EUCAST sensitivity cutoffs are indicated by the bars. Biological triplicates of phage assay are represented individually and bars indicate both neat phage (>2 × 10^2^ PFU/mL) and dilute phage (>2 × 10^5^ PFU/mL) cutoffs. (**a**–**d**) Patients G, K, M and N were colonized with the LES. All strains qualified as MDR based on antibiograms and 14/18 (78%) LES isolates showed some lytic activity to phage using the lower sensitivity cutoff. Using the higher sensitivity cutoff of 10^3^ serial dilutions, 9/18 (50%) showed sensitivity to phage. (**a**) Patient G had isolates that are broadly resistant to antibiotics-sensitive to tobramycin at 2 and to meropenem at 1 time points, respectively, but broadly sensitive to phage at all time points; (**b**) Patient K was colonized by a different strain at the first time point, but all subsequent isolates were LES with lytic activity to phage; (**c**) Patient M’s LES strain was sensitive only to piperacillin-tazobactam but sensitive to phage at all time points; and (**d**) Patient N was colonized with LES resistant to all antibiotics tested. Only 1 of this patient’s 5 isolates showed lytic activity to the neat phage cocktail and no isolates were deemed sensitive using the higher cutoff.

In line with the published literature [31], antimicrobial resistance was commonly seen for the LES strain (Figure 2); all isolates qualified as MDR based on their antibiogram, although there were isolates demonstrating sensitivity to several antibiotics (one LES isolate in patient G was sensitive to ciprofloxacin, tobramycin and meropenem with one other sensitive to ciprofloxacin, tobramycin and amikacin; three LES isolates in patient K were sensitive to a both aztreonam and piperacillin-tazobactam; and three LES isolates in patient M were sensitive to piperacillin-tazobactam). However, every isolate from three patients and one from the fourth were susceptible to the neat bacteriophage cocktail and lytic activity was retained at 10^−3^ dilution (PFU of 2 × 10^5^ PFU/mL) in 9 of 17 LES isolates (Figure 2a–d).

## 4. Discussion

Bacteriophages are being re-explored as potential therapies, an avenue supported by our data demonstrating that the majority of these strains possessed sensitivity to this four-phage cocktail, even those which were MDR. However, we showed variability in both antibiotic and bacteriophage susceptibility in chronic Pa-infected CF patients over 8–12 months. To our knowledge, this is the first time such variability has been explored with bacteriophages and compared with antibiotic susceptibility. Our data suggest that individual patient phage susceptibility will need to be assessed at a timepoint close to the start of phage therapy.

We demonstrated high levels of MDR Pa in our group, roughly half of all isolates being classified as MDR. This is higher than rates reported elsewhere and is likely explained by our selection criteria. Patients were selected on the basis of not only having chronic infection, but on the availability of at least four Pa-positive samples spread throughout one year, which biases towards sicker patients being seen more frequently in hospital. This group is, therefore, one with a high burden of selective pressure in the form of antimicrobial therapy (oral, intravenous, nebulized), a major risk factor for AMR. However, variability data for phages were similar in the MDR and non-MDR cohorts, suggesting our group may be representative of CF populations more widely. We were encouraged by the rates of phage susceptibility in this difficult-to-treat subgroup.

The majority (87.5%) of our cohort had the same VNTR type isolated from each of their samples, providing the cleanest group in which to assess variability to both antimicrobial drugs and phages. We did not set out to compare the rates of variability with the two therapies, as unlike antibiotics, there are no established breakpoints for bacteriophage susceptibility testing. We used the phage cocktail available to us at its highest concentration (4 × 10^9^ PFU/mL) and also assessed lysis to more dilute (log_10_) concentrations. Pragmatically, we have here compared strains based on lysis to both neat and >10^−3^ dilutions in our in vitro analysis, although neither is evidence-based. As would be expected, the lower concentration cut-off led to more patients falling into the variable group (83%) than did the higher concentration; in the latter, roughly half were stably susceptible to phage. In the absence of clinical studies, it is not possible for us to infer a minimum effective dose (MED) for phage. The phages used in this study, however, were used to treat an MDR Pa infection in a CF patient at a concentration of 4 × 10^9^ PFU/mL with a lack of adverse events and clinical resolution of infection [21]. Randomized, placebo-controlled studies are required to determine the cut-off points for phage susceptibility testing and whether an MED can be established. Furthermore, there are pressing questions about whether the co-administration of antibiotics and bacteriophages would result in antagonism or synergy, with comprehensive reviews on this topic which summarise the experimental data [32,33]. This question is especially relevant in the context of CF, where phage therapy will almost certainly be adjunctive to standard antibiotics.

In the majority of cases, the phage susceptibility variability was not the result of different strains at time points tested; as most patients had only a single VNTR type at all time points. Three patients did isolate two distinct strains, an observation which was well-recognised in CF. Whilst only a small group, underpowered for any statistical comparisons, these subjects demonstrated high levels of variability in both antibiotic and phage susceptibility (fourth column, Table 2). Of note, it is unclear whether these patients were co-infected by both strains during the study or whether the later isolate replaced the earlier one. It is well recognized that a sputum sample is generated from only a small portion of the airway tree and substantial geographical heterogeneity in infection and inflammation has previously been documented [34]. It would therefore be possible in a dual-infected patient to culture only one strain on any given occasion. However, in our cohort (i) no patient cultured both their strains at the same time point and (ii) VNTR types were seen either earlier or later in the time course; there was no change from one strain to the other and back again. Since 2010, Public Health England has typed Pa isolates annually from CF patients at RBH and has found that 192 of 1068 patients (18%) harboured more than one strain type (personal communication, Public Health England). Although published rates of infection by more than one strain vary, one Danish study using a different methodology identified 454 distinct strains from 195 patients at 51 different centres over a 30 year period [35].

High rates of variability in antibiotic susceptibility testing have been seen in other studies [36,37], although the majority of these have examined either short time frames or single sampling time points and have not taken the longitudinal approach used in this study. Notably, Foweraker et al. [36], found that there was very poor correlation in antibiotic sensitivity profiles when looking at a single sputum sample—even when testing a single morphotype. It is possible, therefore, that our variability data reflect the same issue, that populations of Pa within the airway possess inherent differences in their susceptibility, even at a single time point. There is also a poor correlation of antibiotic sensitivity testing results and clinical outcomes in CF [16,17]. This well-recognised phenomenon, thought to relate to different modes of growth of bacteria in the CF airway versus on a culture plate/broth, means that CF physicians may be more likely to be guided by clinical response than lab susceptibility testing.

The longitudinal variability in sensitivity could also be the result of antibiotic usage within the patients exerting a selective pressure on their organisms. In our group, variability was greatest for ciprofloxacin and lowest for tobramycin. Both these antibiotics had been administered to our cohort during the study, although design limitations meant that the treatment episodes in individuals could not be aligned with the sampling times. Ciprofloxacin is used in short courses (2–3 weeks) and is well known to cause rapid, often transient, resistance [38]. In contrast, tobramycin was being received on a chronic inhaled basis by 92% of the group, a mode of delivery which may be less likely to cause variability. Usage patterns may therefore, at least in part, explain this variability.

The variability to phages is clearly not related to selective pressure from administered phage and must have another underlying explanation. We considered an intriguing hypothesis that viruses and lysogenic phages within resident bacteria may affect Pa populations in the lung with antiviral defences expressed in subpopulations when particular conditions arise [39]. It has been shown that lysogenic phages infect Pa and incorporate themselves into the host genome as prophages and evidence of prophage DNA in many clinical strains is well documented [40,41,42], bringing to light an ongoing interaction between phages and Pa. Our isolates have not been fully sequenced, a future endeavour which could shed light on this possibility.

However, we cannot rule out the possibility that there are subpopulations within the lung at any time point with variable responses. We did not undertake any analysis (nor are we aware of work elsewhere) of phage susceptibility from specific colonies/colony morphotypes from a single sputum sample as was performed by Foweraker et al. for antibiotics. Our phage sensitivity tests, although performed in biological triplicate, were from stocks of isolates that were obtained from a sweep of the original agar plate, not a single colony; retrieval of a bead is likely to represent this population and the within-sample biological repeat data were tighter (Figure 1 and Figure 2) than might have been expected if multiple response-types were present. Finally, we may simply have been sampling different regions of the lung at various time points, as noted in the discussion of antimicrobial susceptibility.

## 5. Conclusions

The rates of phage-induced bacterial lysis in Pa strains, including those with MDR, provide further support for the assessment of phage therapy for CF. The current study expands our understanding of the longitudinal phenotypic diversity of Pa populations within individual patients and how this translates to varying rates of resistance to antimicrobials in general. In future clinical trials, efficacy read-outs seem intuitively most likely in subjects with susceptible strains, although the lessons of the in vitro–clinical disconnect for antibiotics should not be overlooked. A direct relationship between laboratory susceptibility testing and clinical response may not be assumed and an assessment of this should underpin future trial design.

On the basis of our findings, we suggest (a) isolates tested during screening for eligibility should be from current rather than historical samples, (b) several colonies from an airway sample should be tested and (c) susceptibility should be assessed at more than a single time point. It would perhaps be most informative from both efficacy and safety perspectives to design a trial with enough participants to include patients in whom in vitro testing indicates phage resistance as a control group, albeit phage adaptation in vivo may still be possible.

The recent availability of *cystic fibrosis transmembrane conductance regulator* (CFTR) modulator therapies for people with CF has been heralded as transformative and is likely to change the future of CF disease trajectories [43]. The early life initiation of these drugs may delay or prevent chronic Pa infection, but the impact on infection in patients already chronically infected has been modest and non-durable [44]. The adult CF population, which currently exceeds paediatric age in many parts of the world and is growing, will continue to experience adverse health consequences of chronic Pa infection. New antimicrobial approaches will continue to be an unmet need, even in those receiving CFTR modulator drugs.

## Figures and Tables

**Figure 1 microorganisms-09-00660-f001:**
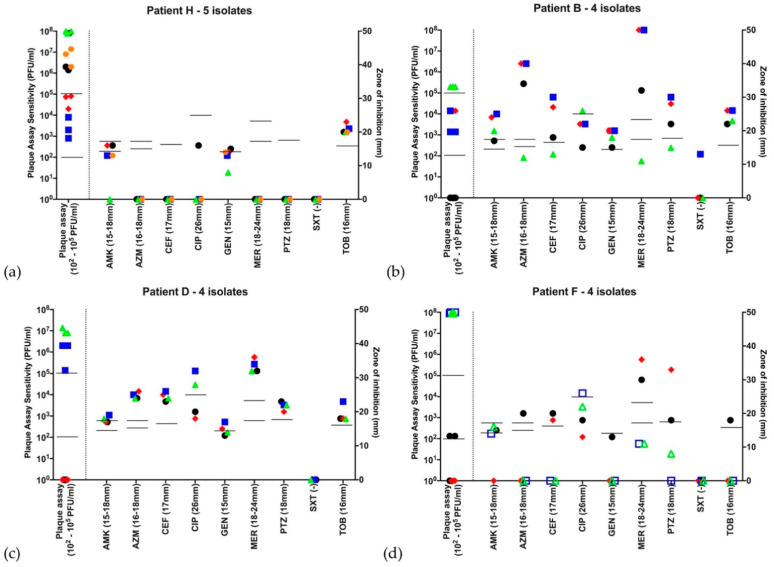
Example antibiograms and bacteriophage sensitivity profiles for 4 different patients chosen to illustrate variability in results. Isolates are graphically represented by different symbols chronologically throughout the year. The first isolate is represented by 
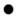
, with subsequent isolates 
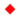
, 
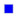
, 
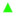
, 
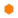
, if there were 5 isolates through the course of 2016. Open symbols represent different strains, identified by VNTR analysis. Abbreviations: AMK: amikacin; AZM: aztreonam; CEF: ceftazidime; CIP: ciprofloxacin; GEN: gentamicin; MER: meropenem; PTZ: piperacillin-tazobactam; SXT: co-trimoxazole (no EUCAST breakpoint); TOB: tobramycin. Disc diffusion assay for antibiotic sensitivity testing was performed once at each time point and EUCAST sensitivity cutoffs are indicated by the bars. Biological triplicates of phage assay are represented individually and bars indicate both neat phage (>2 × 10^2^ PFU/mL) and dilute phage (>2 × 10^5^ PFU/mL) cutoffs. (**a**) Patient H is an example of a patient colonized with MDR Pa, sensitive only to tobramycin and phage at all 5 time points throughout 2016. (**b**) Patient B represents a patient harboring a Pa strain showing high variability to all antibiotics as well as the phage cocktail. Of note, this patient’s first isolate was resistant to the phage cocktail, while subsequent isolates throughout the year showed susceptibility. (**c**) Patient D harbored a Pa strain resistant to the phage cocktail at the first two timepoints in the year, but sensitive throughout the rest of 2016. (**d**) Patient F was one of three from whom two distinct Pa strains were identified. The first two isolates in the year are one strain, while the subsequent two are another. There are distinct antibiograms for the two strains, as well as a distinct pattern of susceptibility/resistance to the phage cocktail.

**Table 1 microorganisms-09-00660-t001:** Characteristics of the patients and isolates.

Patients:	Number	24
	Age (median (range)) in years	31 (14–57)
	Male:female ratio	12:12
	Best FEV_1_ (% predicted median and range) for 2016	62 (23–120)
	Number of isolates per patient (median (range))	4 (3–6)
	Number (%) of subjects with exclusively non-mucoid strains	3 (12.5%)
	Number (%) of subjects with exclusively mucoid strains	6 (25%)
	Number (%) with >1 VNTR type over time period examined	3 (12.5%)
	Subjects (*n*) with shared strains: Liverpool Epidemic Strain *	4
	Cluster A *	2
**Isolates:**	**Number of isolates (total)**	112
	Number with antibiotic and phage sensitivity + VNTR	102
	Number (%) of non-mucoid/ mucoid isolates	44 (43%)/ 58 (57%)
	Number (%) sensitive to neat phage	76 (75%)
	Number (%) sensitive to dilute phage (10^−3^ dilution)	51 (50%)
	Number (%) with resistance to ≥1 antibiotic	88 (86%)
	Number (%) meeting definition of MDR **	51 (50%)
	Number (%) of MDR isolates sensitive to neat phage	40/51 (78%)
	Number (%) of MDR isolates sensitive to dilute phage (10^−3^ dilution)	26/51 (51%)

* [30]. ** Isolates were classified as having multi-drug resistance (MDR) if the disc diffusion assay demonstrated resistance to at least one antibiotic in more than one antibiotic class (β-lactam, aminoglycoside, fluoroquinolone). FEV_1_—forced expiratory volume in 1 s; VNTR—Variable number tandem repeat

## Data Availability

Results of the VNTR typing can be seen in the dendogram in the Appendix A

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
