# Peer review of "Variability in Bacteriophage and Antibiotic Sensitivity in Serial Pseudomonas aeruginosa Isolates from Cystic Fibrosis Airway Cultures over 12 Months"

_microorganisms, 2021, doi:10.3390/microorganisms9030660_

Round 1

Reviewer 1 Report

The manuscript entitled „Variability in bacteriophage and antibiotic sensitivity in serial Pseudomonas aeruginosa isolates from cystic fibrosis airway cultures over 12 months” presents a very important research area, that describes phages therapy of bacteria from MDR group, in this case, Pseudomonas aeruginosa. It is well known that this human pathogen is very resistant to many antibiotics, thus it is necessary to investigate other methods that might help to fight it. One such method/therapy is to use bacteriophages that might “work” in synergy with antibiotics, especially for patients with CF.

In my opinion, the article is very well written and all aspects of the methodology are clear enough.  As the authors wrote, there are necessary additional analyses in this area, however, the results presented in this manuscript are a good start for it.

I am recommended this article for publication in the form that it is right now.

Author Response

Many thanks for your review of our study. We made some small changes on the basis of comments from some of the reviewers (including information on the number of technical and biological replicates), but the manuscript remains largely unchanged as per your suggestions. 

We hope this this study will provide the basis for other research as well as start a foundation for clinical trials involving phage. 

All the best and thanks again for your time,

Isaac Martin 

on behalf of Jane Davies, Eric Alton, Sandra Morales and Dervla Kenna

Reviewer 2 Report

Martin et al., presents an interesting and important study comparing bacteriophage and antimicrobial sensitivity of P. aeruginosa isolates in CF patients. With antimicrobial resistance being frequently observed in P. aeruginosa, particularly in chronically-infected CF-patients, investigations into alternative therapies are of critical importance. Authors did a thorough job comparing conventional antimicrobial treatment to phages, providing an important reference point. Including equal number of male and female patients is an example of careful considerations of research design in this manuscript. The finding that % MDR strains sensitive to neat phage cocktail is similar to that of drug-sensitive P. aeruginosa, is promising and sets foundation for the future studies.

Figs 1-2: number of biological and technical replicates should be specified in the legend. Some symbols are missing in the legend.

Line 190-191: This sentence: “And present separately if you agree as above.” seems to be a comment between authors that made it into the document.

Author Response

Many thanks for your review of our manuscript. 

We have made the suggested changes, as per your suggestions, which we have outlined below:

(1) Figs 1-2: number of biological and technical replicates should be specified in the legend. Some symbols are missing in the legend.

  • We have changed this in the final manuscript. Thank your for drawing our attention to this. 

(2) Line 190-191: This sentence: “And present separately if you agree as above.” seems to be a comment between authors that made it into the document.

  • Yes, included in error. 

We hope that this study contributes to the growing body of knowledge on phage therapy and also provides a small foundation for the development of clinical trials involving phage. 

Many thanks, once again, for your time and effort on this,

Isaac Martin

On behalf of Dervla Kenna, Sandra Morales, Eric Alton and Jane Davies